# Soil Microbial Community Varied with Vegetation Types on a Small Regional Scale of the Qilian Mountains

Wen Zhao, Yali Yin, Shixiong Li *, Jingjing Liu, Yiling Dong and Shifeng Su

Qinghai Academy of Animal and Veterinary Science, Qinghai University, Xining 810016, China;
zhaowen19951020@163.com (W.Z.); yyl0909@163.com (Y.Y.); 18409498491@163.com (J.L.);
yilngdong@163.com (Y.D.); sufeng1018@163.com (S.S.)
* Correspondence: shixionglee@hotmail.com

**Abstract:** Clarifying the response of soil microbial communities to the change of different vegetation types on a small regional scale is of great significance for understanding the sustainability of grassland development. However, the distribution patterns and driving factors of the microbial community are not well understood in the Qilian Mountains. Therefore, we characterized and compared the soil microbial communities underlying the four vegetation types in a national natural reserve (reseeded grassland, swamp meadow, steppe meadow, and cultivated grassland) using high-throughput sequencing of the 16S rRNA and ITS. Meanwhile, the plant community and soil physicochemical characteristics were also determined. The results showed that bacterial and fungal communities in all vegetation types had the same dominant species, but the relative abundance differed substantially, which caused significant spatial heterogeneities on the small regional scale. Specifically, bacteria showed higher variability among different vegetation types than fungi, among which the bacterial and fungal communities were more sensitive to the changes in soil than to plant characteristics. Furthermore, soil organic carbon affected the widest portion of the microbial community, nitrate-nitrogen was the main factor affecting bacteria, and aboveground plant biomass was the main factor affecting fungi. Collectively, these results demonstrate the value of considering multiple small regional spatial scales when studying the relationship between the soil microbial community and environmental characteristics. Our study may have important implications for grassland management following natural disturbances or human alterations.

**Keywords:** Qilian Mountains region; small regional scale; different vegetation type; soil microbial; environmental factors

## 1. Introduction

Soil microbes play a pivotal role in the functioning of terrestrial ecosystems, which is considered an important indicator that monitors the succession of ecosystems [1]. Therefore, it is indispensable to improve the knowledge of the microbial community, especially regional microbial patterns [2]. Although encouraging progress has recently been made on the horizontal distribution of microbial communities [3,4], we still lack an understanding of the different ecological attributes of soil microbial communities, such as the spatial distribution of microbial communities and the driving factors of maintaining diversity at the small regional scale [5]. This is partially due to methodological constraints, and any investigation of the soil microbial community encounters the problem that the substrate is highly heterogeneous on large scales, both horizontally and vertically [6]. The relatively large number of uncontrollable factors in analyzing microbial regional variation cannot mitigate the influence of spatial distance, which causes differences in both the structure and function of soil microbial communities.

The distribution pattern and driving mechanism of the soil microbial community are core topics of microbial ecology research [7]. Previous studies have confirmed that microbial composition and diversity are regularly distributed in space along with changes

in biological and abiotic factors, such as climate and physicochemical properties [8–10]. At present, many kinds of research pay more attention to the scale of large regions. For example, soil pH is considered as the most important abiotic factor, which drives the diversity, composition, and biomass of microbial communities on a global scale [4]. However, the study found that pH had no significant correlation with microorganisms in arid areas at a small regional scale [11]. Therefore, the effects of abiotic and biotic factors on soil microorganisms may vary with the regional scale of the research. In summary, understanding microbial community composition and diversity on the same small scale is an important first step in evaluating microbial community structure–function relationships [12]. First, exploring the interaction relationship of microorganisms and environmental factors at a small spatial scale will help answer the basic scientific questions of different ecologies. Second, understanding the main driving mechanism of microbial differences in different ecosystems is important for designing effective management and conservation strategies.

Alpine grassland vegetation composition varies greatly across short distances due to heterogeneities in topography and hydrological conditions across the landscape [13]. Changes in vegetation types can alter plant community composition and soil characteristics, and these alterations can, in turn, affect soil microbial communities due to changing composition and diversity [14,15]. Both the quality and quantity of aboveground litter and belowground roots supplied to soil microorganisms differ among vegetation types [16]. On the other hand, changes in soil properties, such as pH, moisture, clay, C, N, and phosphorus availability, under different vegetation types, have significant impacts on soil microbial communities [17,18]. Nottingham revealed that ecosystems with strong plant community adaptability to changing resources will have homeostatic microbial communities with relatively low microbial resource costs because plants reduce variance in resource stoichiometry [19]. For example, Xue et al. explored different grassland ecosystem types spanning 2121 km across the Tibetan Plateau and found that drought index is the main factor affecting microbial communities [20]. Che et al. demonstrate that the distribution of the microbial community regarding nitrogen fixation on the Tibetan Plateau was mainly affected by soil water content and nutrient availability [21]. Aside from other abiotic factors, the composition of microbial communities appears to depend on the prevailing and dominant available source of organic carbon (C) among different habitats [22]. In addition, a recent meta-analysis indicated that the cross-habitat distribution pattern of bacteria was more strongly driven by habitat type on a global scale [23]. In summary, the distribution of the microbial community and driving factors at a small scale are not well understood, despite the plentiful existence of global and large scale studies.

To comprehensively understand the differences in the soil microbial community in alpine grasslands at small regional scales, the microbial ecology among different vegetation types was investigated on the east side of the Qilian Mountains, which is an ecologically fragile area at the convergence of Qinghai-Tibet, Mongolia-Xinjiang Plateau, and the Loess Plateau [24]. Previous studies have evaluated the interactions among soil, plant, and microbial communities on a large regional scale in the Qinghai-Tibet Plateau [25]. However, the microbial responses to different vegetation types on a small regional scale are not well understood. In this study, we used high-throughput sequencing technology to analyze the changes in the soil microbial diversity and community among four vegetation types (no more than 10 km apart). Based on previous studies related to soil microbes in different vegetation types on a large regional scale, the goal of this research was to (1) clarify how the composition and diversity of soil bacterial and fungal communities change among different grassland types; (2) explore the influence of vegetation and soil on microorganisms; (3) determine the key factors that affect the changes in the microbial community.

## 2. Methods and Materials

### 2.1. Study Area

The study was conducted at the Qilian Mountain Nature Reserve on alpine grassland, located in a typical alpine meadow, Qilian County, Mule town (37°57′ 36″ N, 100°19′48″

E; 3487 m above sea level). The region is characterized by a continental plateau climate, with a mean annual temperature of −1.7 °C, and the annual precipitation between June and September is approximately 614.8 mm. The region's highest and lowest recorded temperatures were in January (−14.8 °C) and July (9.8 °C), respectively. The plant growing period is approximately 150 d, and there is no absolute frost-free period throughout the year [26].

This area is rich in grassland types, and some degraded grasslands have been converted to cultivated or reseeded grasslands. The cultivated grassland was established on the severely degraded alpine meadow for 4 years, mainly with *Poa pratensis* cv. Qinghai. The reseeded grassland was dominated by *E. nutans*, which was reseeded on the moderately degraded alpine meadows for 2 years by a reseeding/sowing combined operation (Figure 1). The swamp meadow and steppe meadow are natural grasslands with the dominant species *Kobresia humilis* and *Elymus nutans*, respectively. Each vegetation type was used for grazing with the same intensity.

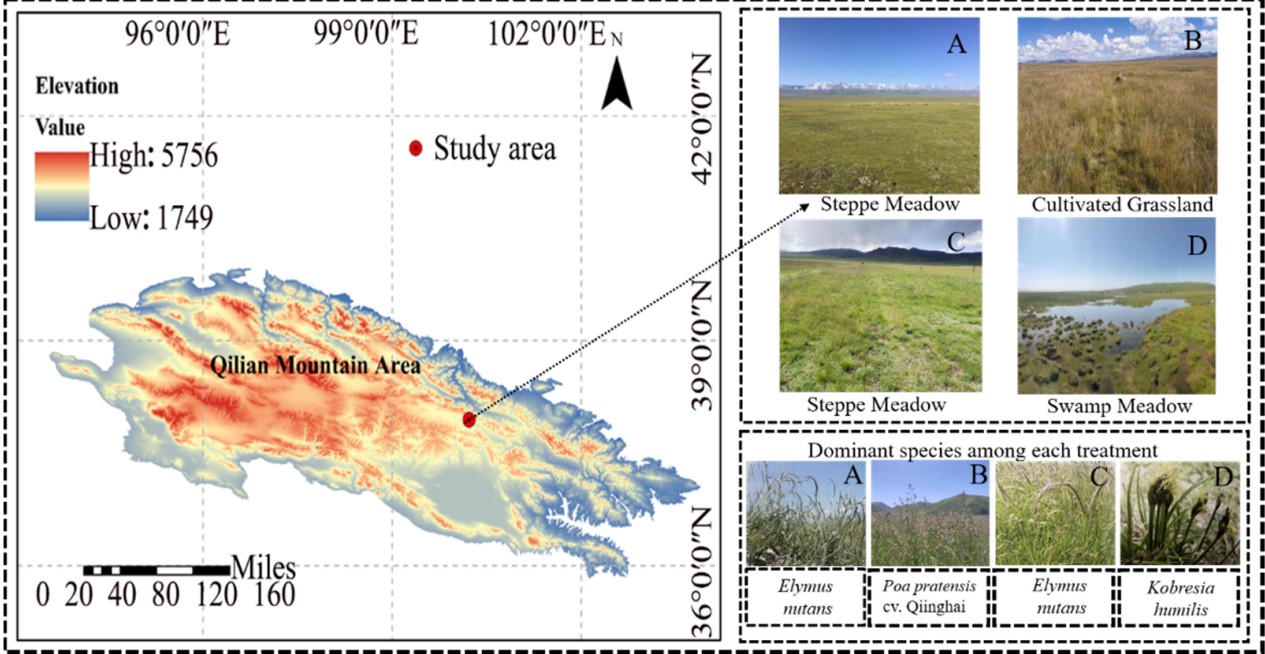

**Figure 1.** Spatial distribution of sampling sites across different grassland types in the small regional scale of the Qilian Mountains region; the photos of the grasslands are (**A**) steppe meadow with dominant *Elymus nutans*, (**B**) cultivated grassland with dominant *Poa pratensis* cv. Qinghai, (**C**) reseeded grassland with dominant *Elymus nutans*, and (**D**) swamp meadow with dominant *Kobresia capillifolia*.

## 2.2. Experimental Design, Plant Investigation, and Soil Sampling

We surveyed the vegetation community at the site, and located four similar, well-separated (500–1000 m apart) patches of each vegetation type (August 2019). This study design with replicate plots (rather than replicate plots within a patch) avoids pseudoreplication and renders our results appropriate for landscape-level extrapolation. In each patch, species diversity, abundance, total coverage, and height were investigated in four 50 cm × 50 cm quadrats. All aboveground plant parts were collected in each quadrant as the aboveground biomass, which was determined by oven-drying plant samples at 65 °C for 48 h in the laboratory to a constant weight. At the same time, root samples from a 0–15 cm depth were collected separately using a root drill with a diameter of 7 cm, after which four root drill samples from each plot were mixed to give one composite root sample [27].

Soil samples were collected from quadrats: 8 random soil samples were collected from the 0 to 15 cm soil layer in each replicate plot using a soil-drilling sampler (7 cm inner diameter) and combined into 1 replicate sample; in total, 4 soil samples were obtained from

each vegetation type. All soil samples were passed through a 2 mm sieve to remove other materials. Soil samples were immediately sent back to the laboratory in a cooler. Samples were divided into three parts to determine the chemical and physical properties of the soil, along with the soil microorganisms. One part of the soil sample was naturally dried in the shade and then sieved through 1 mm mesh for soil chemical analysis, while another part was stored at $-80\ ^{\circ}$C for high-throughput gene detection [28]. A third portion was used in a water-stable aggregate analysis.

### 2.3. Measurement of Soil Physicochemical Properties

The physicochemical properties of the soil samples were analyzed within one month. The soil bulk density (0–15 cm) was measured using the metal-ring method and oven-dried at 105 $^{\circ}$C for 24 h. Soil pH was determined in a water–soil suspension with a volume ratio of 1:5 [29]. Water-stable aggregates were generated by using nested sieves (2 mm, 0.05 mm, and 0.02 mm), according to the procedure described by [30]. The contents of soil organic carbon (SOC) and total nitrogen (TN) were determined by a C and N analyzer (Elementar, Langenselbold, Germany). Soil available nitrogen was extracted with 1 M KCl, and the filtrates were analyzed for $NH_4^+$-N and $NO_3^-$-N using a colorimetric method analyzer (CleverChem200+, Hamburg, Germany) [31].

### 2.4. Bioinformatics Analyses

The soil microbial community composition and diversities were determined using high-throughput gene detection techniques. Soil samples stored at $-80\ ^{\circ}$C were transported with dry ice to the Guangzhou Genedenovo Biological Technology Center for analysis (Illumine 2500 250 PE, USA). For each sample, the total genomic DNA was extracted from 0.5 g of soil using a HiPure Soil DNA Mini Kit (Magen, Guangzhou, China). A NanoDrop device (NanoDrop 2000, Thermo Fisher, Waltham, MA, USA) was used to detect the DNA quality. The 16S rDNA V3-V4 region of the ribosomal RNA gene was amplified by PCR (94 $^{\circ}$C for 2 min, followed by 30 cycles at 98 $^{\circ}$C for 10 s, 62 $^{\circ}$C for 30 s, and 68 $^{\circ}$C for 30 s, and a final extension at 68 $^{\circ}$C for 5 min) using primers 341F (CCTACGGGNGGCWGCAG) and 806R (GGACTACHVGGGTATCTAAT) [32]. ITS rRNA was detected in the ITS2 region, and the primer sequences were KYO 2F (GATGAAGAACGYAGYAA) and ITS4R (TC-CTCCGCTTATTGATATG). PCRs were performed in a triplicate 50 µL mixture containing 5 µL of 10 × KOD buffer, 5 µL of 2 mM dNTPs, 3 µL of 25 mM MgSO4, 1.5 µL of each primer (10 µM), 1 µL of KOD polymerase, and 100 ng of template DNA [33]. Bacterial and fungal $\alpha$- diversity indices, including the Chao 1, Simpson, and Shannon indices, were calculated at the OTU level. The Sobs index indicates the number of OTU actually measured. Alpha-diversity metrics (including the Chao1 richness estimator and the Shannon diversity index) were estimated for each sample using the diversity plugin in QIIME2 software. Redundancy analysis (RDA) was conducted at the phylum level.

### 2.5. Statistical Analyses

The analysis of variance (ANOVA) was used to determine the significance of differences in the microbial diversity index, soil physicochemical properties, and plant diversity among different vegetation types. The significant differences were determined at the 95% confidence level. PCoA is a dimensionality reduction analysis of the microbial community based on the Bray–Curtis distance to evaluate the interpretation degree of each coordinate axis for the bacterial community structure by percentage. Generally, it is reasonable for the sum of PCoA1 and PCoA2 to reach more than 50%. A low interpretation degree can be acceptable in a complex community. The grouping test is usually combined with a PCoA scatter map, which visually presents the grouping characteristics of samples, while the grouping test is used to judge whether the community differences are significant. In addition, we used the ADONIS function of the vegan package in R 4.0.3 (Ross Ihaka and Robert Gentlemen, Auckland, New Zealand) to test the significance of the separation between successional stages. ANOSIM was used to detect differences within and between

groups. The linear discriminant analysis (LDA) effect size (LEfSe) method was used to assess high-dimensional microbial taxa and identify the taxonomically different microbial vegetation types. The steps of LEfSe analysis are as follows: the Kruskal–Wallis rank-sum test (a commonly used test method for multiple groups) was conducted between all groups, and then the selected differential species between two groups were compared by the Wilcoxon rank-sum test (a commonly used test method for two groups). Finally, the differences were selected by using LDA, and the results were sorted by mapping to obtain the evolutionary branches. The significant environmental factors in the RDA were selected by the envfit test, and the best model was selected by the 'mod' function of the vegan package [34]. Redundancy analysis (RDA) was used to elucidate the relationships between the community and environmental properties by using the R vegan package, and the relationships were examined using the Monte Carlo permutation. Furthermore, VPA (variance partitioning analysis) based on the abundance of species and environmental factors was used to analyze the interpretation of the total variation in species distribution using each group of environmental factor variables. We also calculated the Bray–Curtis distance matrix between samples based on the species abundance and the environmental factor, and used the mantel test to analyze the correlation between the two distance matrices, using the R vegan package for analysis.

## 3. Results

### 3.1. Microbial Community Composition and Diversity

Bacterial and fungal communities were characterized by sequencing the V3–V4 hypervariable region of the 16S rRNA gene and the ITS region, respectively. The soil sample was harvested from 4 grassland types. A total of 10 bacterial phyla were detected in 16 soil samples (51 genera and 5316 OTUs). The relative abundance of the dominant bacterial phyla was greater than 5% in the 4 grassland types, which were Proteobacteria, Acidobacteria, Planctomycetes, and Actinobacteria. (Figure 2A). The highest relative abundance of Proteobacteria was observed in SW, which was significantly higher than that in ST and RG. Moreover, ST had a higher abundance of Planctomycetes and Actinobacteria than CG and SW. The relative abundance of Acidobacteria in RG was higher than that in the other grassland types.

Additionally, 7 phyla, 11 genera, and 1544 OTUs were also identified in the soil fungal community, which primarily comprised members of the phyla Ascomycota, Basidiomycota, and Mortierellomycota, among which Ascomycota (59.99–80.15%) was a dominant phylum for each grassland type (Figure 2C). The highest relative abundance was observed in CG (Figure 2D). By using LEfSe analysis, we found that changing the vegetation significantly altered the bacterial and fungal communities. Specifically, at the phylum, genus, and species levels, some bacterial and fungal groups were significantly enriched in the different grassland types. Bacteria were mainly enriched in SW. However, most fungal groups were enriched in CG. More detailed information is shown in Figure S1.

As illustrated by the PcOA analysis based on Bray–Curtis dissimilarity, at the OTU level, PCoA1, and PCoA2 contributed 48.89% and 16.36%, respectively (Figure 2C). Comparatively, for fungal structure, PCoA1 and PCoA2 contributed 25.24% and 15.81% of the total variation, respectively (Figure 3B). PCoA analysis suggested that the soil bacterial and fungal structures among different vegetation types were clearly separated from each other. Simultaneously, PERMANOVA was applied to reveal differences in the soil microbial beta diversity (Table S1), which indicated that there was a significant difference among the four grassland types ($R2 = 0.7$, $p < 0.01$). Moreover, there were also significant differences in microbial beta diversity among the other grasslands ($p < 0.05$).

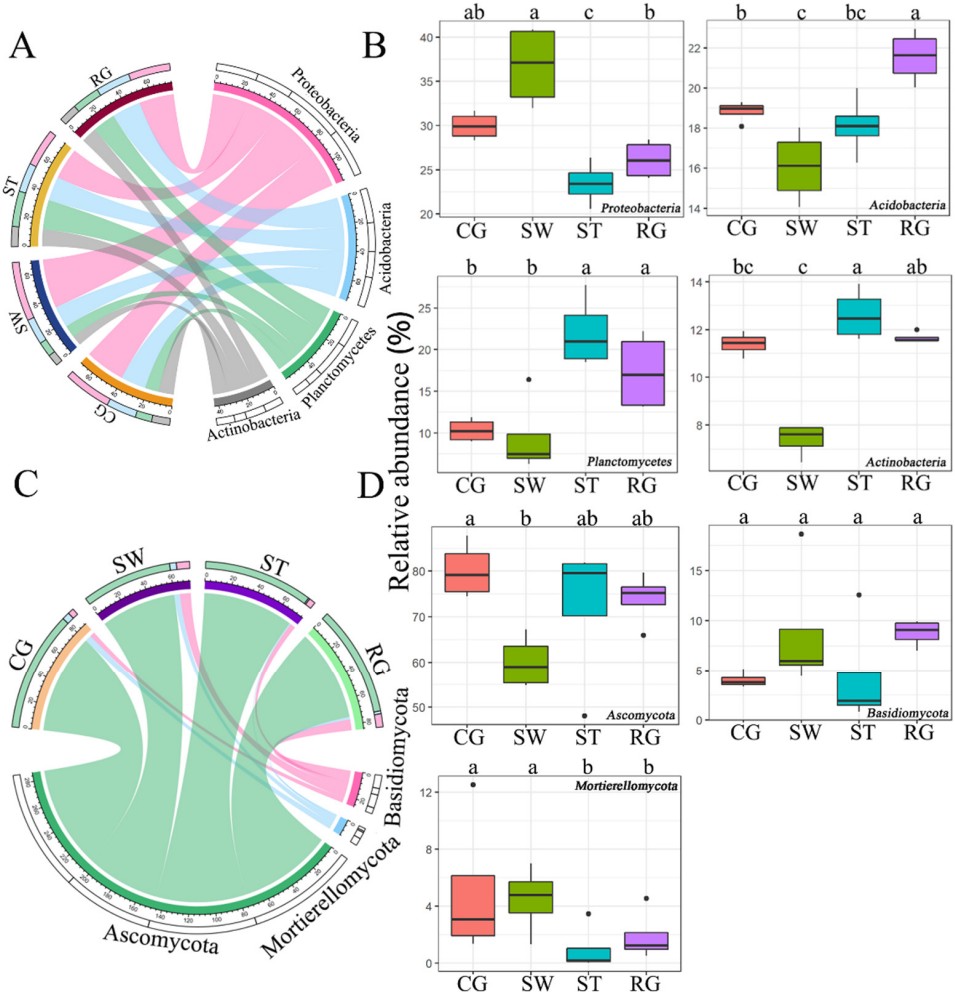

**Figure 2.** Dominant phylum of bacteria and fungi in the soil samples among the four different grassland types. (**A**,**B**) A Circos diagram was used to dynamically show the abundance composition of species among each grassland type at phylum classification levels in soil bacterial and fungal communities. (**C**,**D**) The ANOVA of the relative abundance of each dominant species. CG, SW, ST, and RG indicate cultivated grassland, swamp meadow, steppe meadow, and reseeded grassland, respectively. Different letters from figures indicate significant differences ($p < 0.05$) among different grassland types.

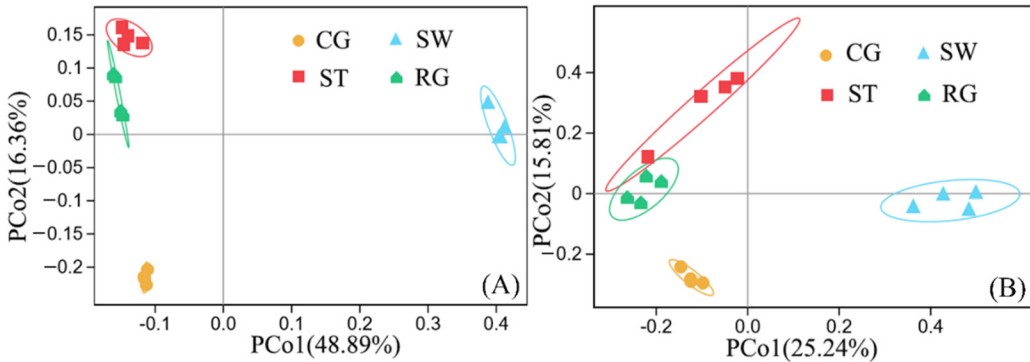

**Figure 3.** Principal component analysis of bacteria and fungi in the soil samples among the four different grassland types at OTUs level; (**A**) bacteria; (**B**) fungi. CG, SW, ST, and RG indicate cultivated grassland, swamp meadow, steppe meadow, and reseeded grassland, respectively.

Similarly, with the change of grassland types, bacterial and fungal diversity also changed significantly (Figure 4). Specifically, the Sobs and diversity index of the soil bacterial community were higher than those of the fungal community. The results also indicated that using Sobs, Shannon, or Chao 1 indices, each grassland type showed the highest levels of ST and the lowest of SW for bacteria. More information is shown Sobs (Figure 4A), Shannon (Figure 4B), or Chao 1 (Figure 4C). Moreover, the Sobs index and diversity of the Shannon and Chao 1 indices in CG were greater than those in the other grassland types for fungi. Similar to bacteria, the lowest fungal diversity was observed in SW, except for the Shannon index.

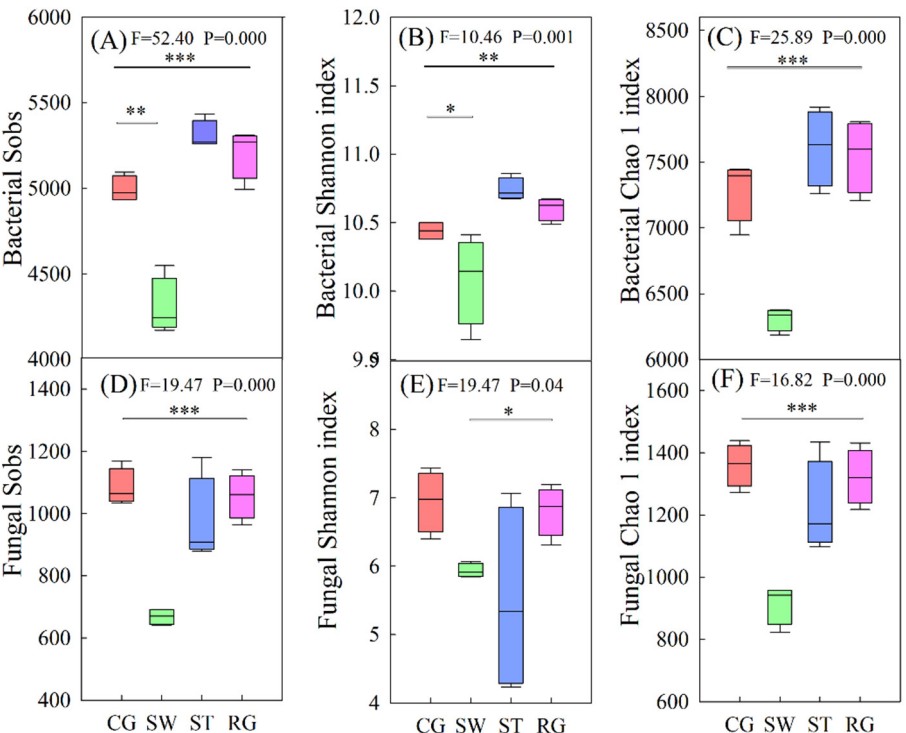

**Figure 4.** Changes in alpha diversity of bacteria and fungi among different vegetation types: Sobs (**A**), Shannon index (**B**), and Chao 1 index (**C**) of soil bacteria. Sobs (**D**), Shannon (**E**), and Chao 1 index (**F**) of soil fungi. CG, SW, ST, and RG indicate cultivated grassland, swamp meadow, steppe meadow, and reseeded grassland, respectively. Asterisks indicate statistical significance (*** $p < 0.001$; ** $p < 0.01$; * $p < 0.05$).

### 3.2. Differences in Soil Physicochemical Characteristics among Different Vegetation Types

Soil physicochemical characteristics varied greatly among each grassland type, which reflected the high heterogeneity of soil characteristics among different grassland types. For all measured soil parameters, significant differences were observed within the 0-10 cm soil layer among the four grassland types (Table 1). At the SW site, the $NH_4^+$-N, $NO_3^-$-N, SOC, SWC, TN, and clay were significantly higher than those at the other grassland types ($p < 0.05$). Compared with SW and ST, the soil sand and silt in CG were significantly increased ($p < 0.05$). The content of TP in the SW and ST was significantly higher than that in the RG and CG, but the pH showed a different trend. Furthermore, the BD was reduced in RG and CG, compared with SW and ST. In addition, we also noticed that most soil characteristics, except TP, BD, and silt, were significantly correlated with the soil bacterial community ($p = 0.01$ to 0.033, r = 0.390 to 0.946) based on the envfit test. However, the soil bulk density, soil water content, and silt content had no significant association with the soil fungal communities.

**Table 1.** Characteristics of the soil among different grassland types.

| Grassland Types | RG | SW | ST | CG | Bacteria | | Fungi | |
|---|---|---|---|---|---|---|---|---|
| | | | | | r | p | r | p |
| $NH_4^+$-N (mg/kg$^{-1}$) [a] | 1.56 ± 0.37 c | 3.06 ± 1.37 a | 1.95 ± 1.34 b | 1.45 ± 0.57 c | 0.4578 | 0.019 | 0.5089 | 0.009 |
| $NO_3^-$-N (mg/kg$^{-1}$) [a] | 1.94 ± 0.36 b | 5.79 ± 1.95 a | 2.21 ± 0.52 b | 1.74 ± 0.31 b | 0.8641 | 0.001 | 0.6676 | 0.003 |
| SOC (g/kg$^{-1}$) [a] | 19.45 ± 1.24 c | 172.68 ± 19.84 a | 41.80 ± 5.26 b | 28.51 ± 6.22 bc | 0.9426 | 0.001 | 0.6808 | 0.001 |
| TN (g/kg$^{-1}$) [a] | 1.52 ± 0.49 b | 6.68 ± 1.76 a | 2.62 ± 0.70 b | 2.36 ± 0.50 b | 0.7461 | 0.001 | 0.8318 | 0.001 |
| TP (g/kg$^{-1}$) [a] | 0.61 ± 0.02 c | 1.09 ± 0.06 a | 1.05 ± 0.03 a | 0.91 ± 0.04 b | 0.4985 | 0.019 | 0.5507 | 0.007 |
| pH | 7.96 ± 0.37 b | 7.21 ± 0.23 b | 7.31 ± 0.14 a | 7.93 ± 0.14 a | 0.5772 | 0.004 | 0.4863 | 0.010 |
| BD (g/cm$^3$) [a] | 1.48 ± 0.13 a | 1.05 ± 0.26 b | 1.39 ± 0.12 a | 0.88 ± 0.06 b | 0.0388 | 0.772 | 0.3394 | 0.067 |
| SWC (%) | 13.89 ± 1.69 c | 49.19 ± 2.28 a | 18.34 ± 0.85 b | 16.20 ± 1.69 bc | 0.5582 | 0.005 | 0.2484 | 0.151 |
| Sand (%) [a] | 70.56 ± 11.15 ab | 43.66 ± 7.48 c | 64.93 ± 3.51 b | 79.91 ± 6.85 a | 0.7721 | 0.001 | 0.5185 | 0.010 |
| Silt (%) [a] | 1.26 ± 0.59 ab | 3.35 ± 1.37 c | 6.29 ± 1.84 b | 2.71 ± 0.61 a | 0.2189 | 0.214 | 0.334 | 0.069 |
| Clay (%) [a] | 27.42 ± 9.86 b | 52.99 ± 8.84 a | 30.89 ± 3.33 b | 13.80 ± 1.01c | 0.6240 | 0.003 | 0.5694 | 0.007 |

**Note:** $NH_4^+$-N, $NO_3^-$-N, SOC, TN, TP, pH, BD, SWC, Sand, Silt, Clay represent the abbreviations of soil ammonium nitrogen, soil nitrate-nitrogen, soil total carbon, total nitrogen, pH, bulk density, soil water content, soil sand content, soil silt content, and soil clay content. Values are the means ± SE (n = 4). Lowercase letters indicate significant differences among different grassland types ($p < 0.05$). CG, SW, ST, and RG indicate cultivated grassland, swamp meadow, steppe meadow, and reseeded grassland, respectively. [a] represent determination of dry matter of soil.

### 3.3. Differences in the Plant Community among Different Vegetation Types

　　The vegetation community composition varied significantly among each grassland type, especially in terms of plant diversity and biomass. The highest and lowest H indexes were observed in RG and CG, respectively, which was similar to the E index (Figure 5). The D index changed significantly as well. However, there was no significant difference in the S index among the four grassland types. Furthermore, the plant underground biomass in SW was significantly higher than that in the other treatments. In contrast, the aboveground biomass of SW was significantly lower than that of the others. Mantel testing also showed that there was a significant correlation between plant characteristics and the microbial composition at the phylum level (Figure 6). We used RDA to analyze the explanations of plant and soil characteristics for microbial community variation. Surprisingly, our results show that soil characteristics have more influence on microbial communities than vegetation, whether bacteria or fungi. For the bacterial community, the interaction between plants and soil contributed 51.24% of bacterial community variation, with soil explaining 33.53% of the data variance, which was higher than the result for plants, explaining 7.32% of the data variance (Figure 7A). The interaction between the vegetation and the soil of the soil fungal community was lower than that of the bacterial community, but the soil explanatory degree was still greater than that of the plant factors (Figure 7B).

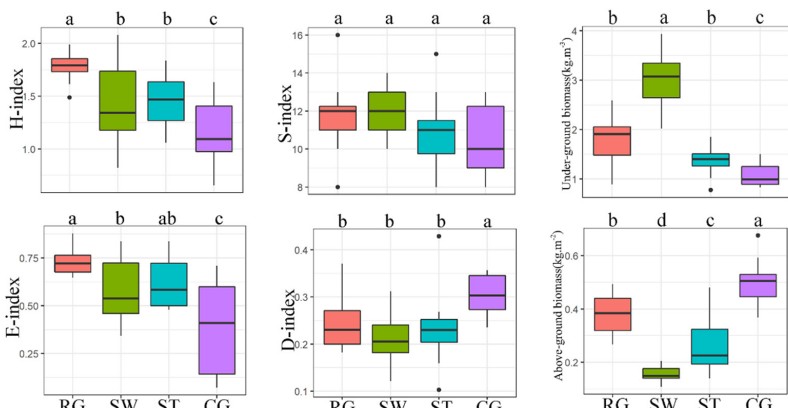

**Figure 5.** Plant diversity and biomass among different vegetation types. Lowercase letters indicate significant differences among different grassland types ($p < 0.05$). CG, SW, ST, and RG indicate cultivated grassland, swamp meadow, steppe meadow, and reseeded grassland, respectively. Different letters indicate significant differences ($p < 0.05$) among different grassland types.

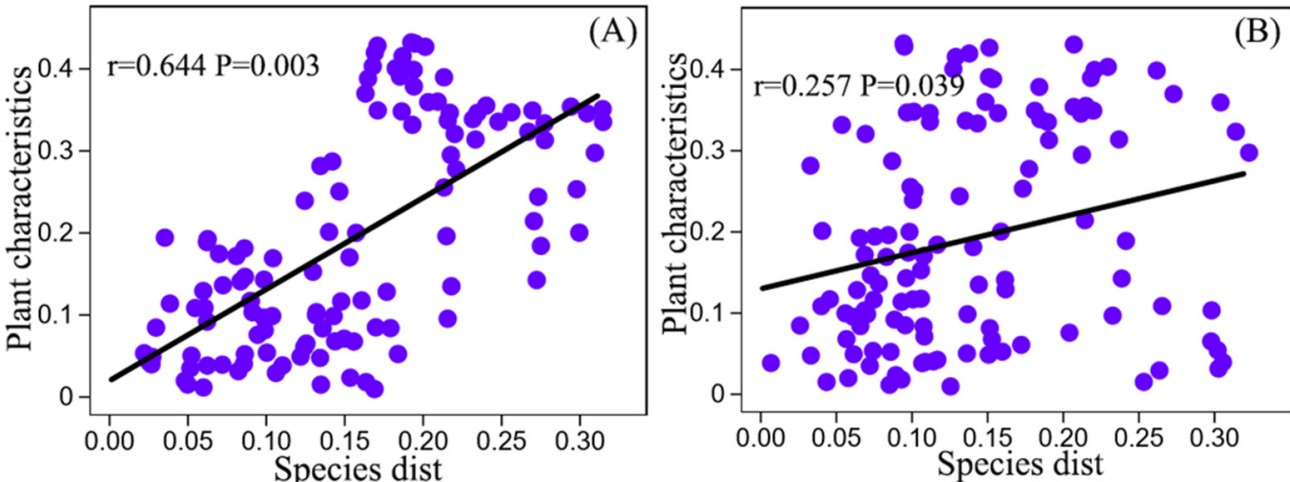

**Figure 6.** Relationships between alpha diversity and plant characteristics were estimated via linear regression analysis based on the UniFrac distance matrix; (**A**) bacteria, (**B**) fungi.

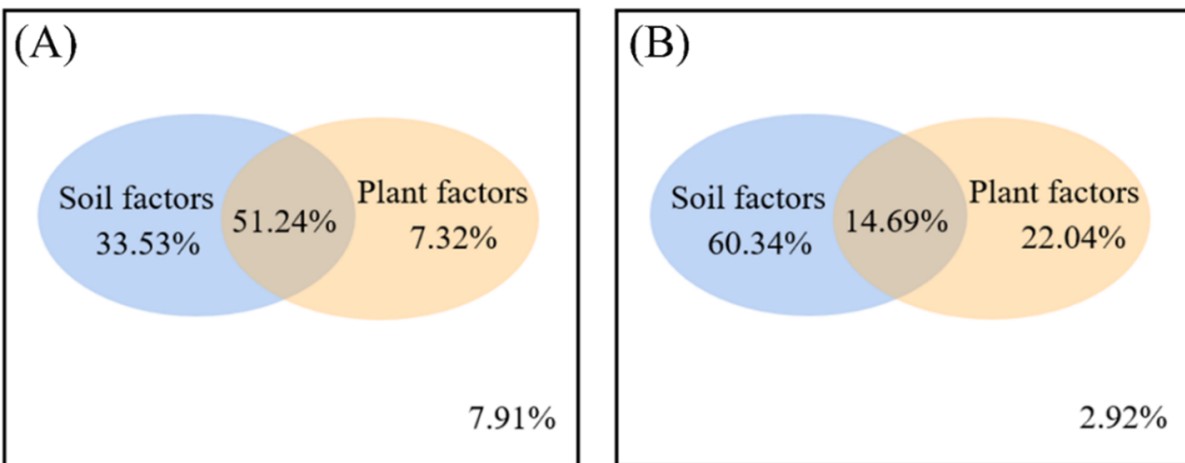

**Figure 7.** The relationships between the soil microbial community with the plant and soil factors, and the interaction between soil and plant explained variation in the bacterial community; (**A**) bacterial community, (**B**) fungal community.

### 3.4. Correlation of Soil Physicochemical Properties and Plant Factors with Microbial Community

Based on the selected variables from the envfit and Mantel test, for soil bacteria, RDA showed eigenvalues of 33.08% and 16.78% for the first and second axes, indicating that these two axes could explain approximately 49.86% of the total variation in soil bacterial communities. Redundancy analysis found that these factors (such as $NH_4^+$-N, TN, SWC, SOC, $NO_3^-$-N, clay, TP, silt) were significantly related to the bacterial community in (SW). Furthermore, most physical properties of soil (sand, pH, BD) could explain the differences in soil bacterial communities among (RG), (ST), and (CG) (Figure 8A). Across the first two canonical axes, RDA also explained 44.73% of the relationship with the fungal community (Figure 8B). In addition, we used VPA (variance partitioning analysis) to analyze the explanation of each environmental variable to the total variation in species distribution. VPA also indicated that $NO_3^-$-N and SOC are the main soil factors that cause the differences in bacteria (contribute > 45%) (Figure 9A). Soil organic carbon is also the dominant factor affecting fungal communities (contribution > 29%). The underground biomass of plants has a greater contribution to bacteria and fungi, but the diversity of plants does not (Figure 9B).

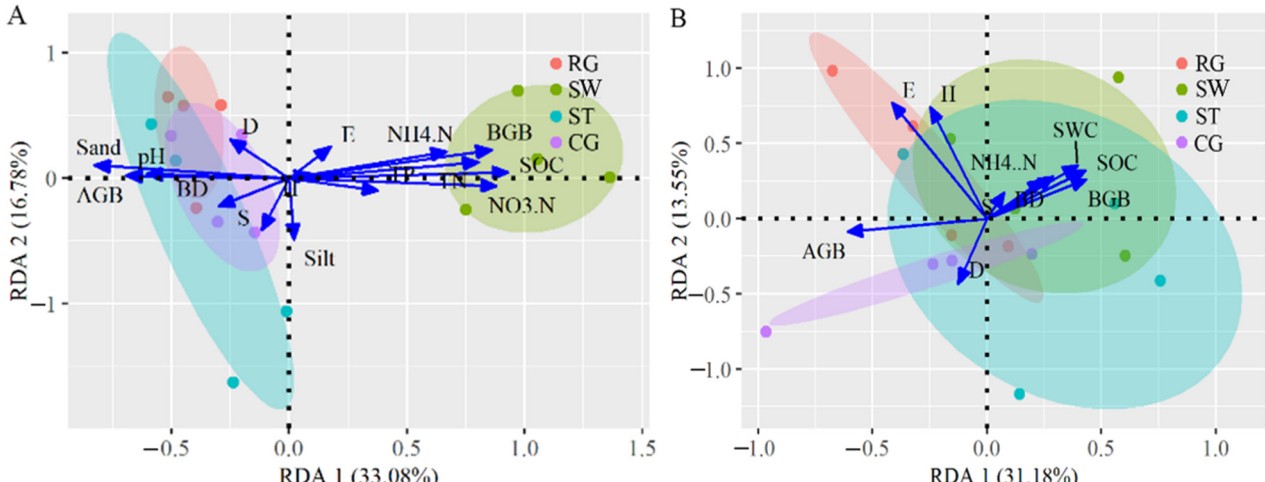

**Figure 8.** Ordination plots of the results from the redundancy analysis (RDA) to identify the relationships between the microbial community and the plant and soil characteristics (blue arrows); (**A**) bacteria, (**B**) fungi. CG, SW, ST, and RG indicate cultivated grassland, swamp meadow, steppe meadow, and reseeded grassland, respectively.

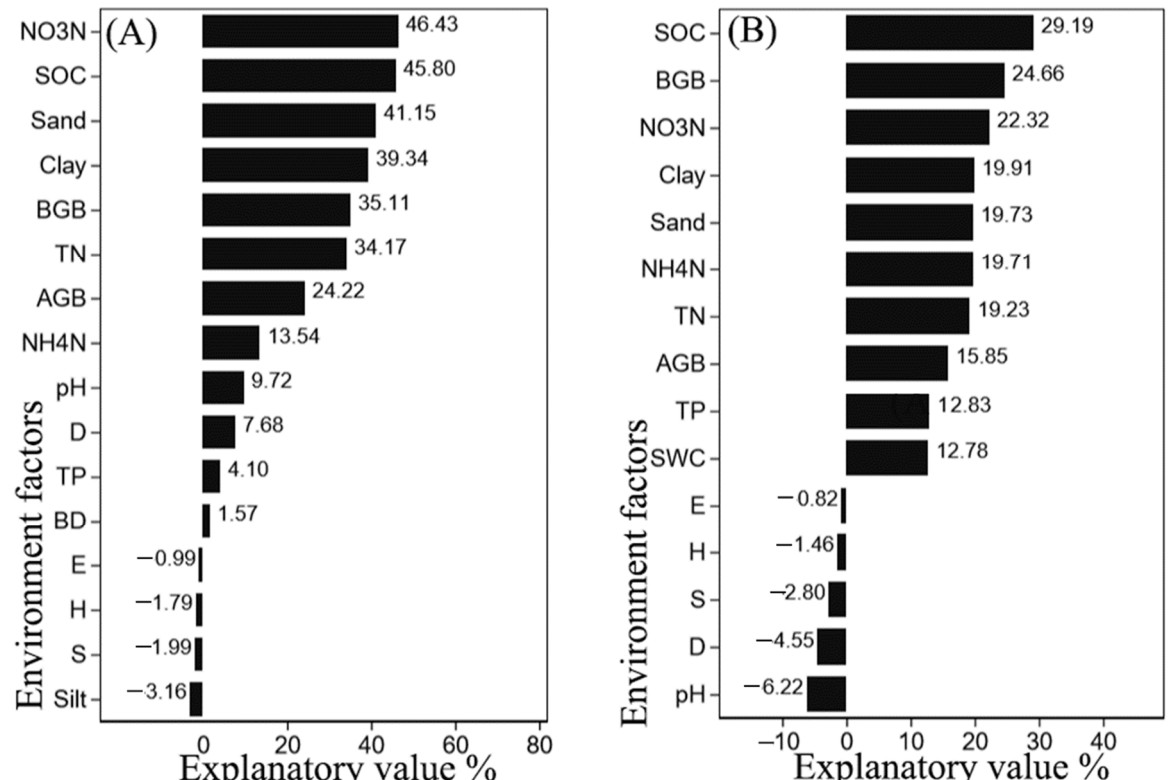

**Figure 9.** The contribution of each environmental factor variable to the total variation of species distribution is analyzed based on variance partitioning analysis (VPA)L: (**A**) bacteria, (**B**): fungi. $NH_4^+$-N, $NO_3^-$-N, SOC, TN, TP, pH, BD, SWC, Sand, Silt, and Clay represent the abbreviations of soil ammonium nitrogen, soil nitrate-nitrogen, soil total carbon, total nitrogen, pH, bulk density, soil water content, soil sand content, soil silt content, and soil clay content. H, D, S, and E represent the abbreviations of vegetation community according to the Shannon–Wiener, Dominance, Richness, and Evenness diversity index. AGB and BGB indicate the plant aboveground biomass and underground biomass.

## 4. Discussion

Our study of the full range of ecosystems along a typical alpine grassland on a small regional scale demonstrates that although the composition of the dominant soil bacteria at the phylum level was similar, their relative abundances differed substantially and consistently among different vegetation types (Figure 2A,B). In addition, the dominant fungi were common across all of the vegetation types. Our study provides strong evidence that soil bacterial and fungal communities vary significantly with the change in habit, although they are on the same regional scale, which is consistent with the findings of previous studies in the alpine meadow of the Qinghai-Tibet Plateau and boreal forests [35,36]. Interestingly, we also notice that the dominant phylum of fungi Ascomycota was significantly decreased in the swamp meadow. Concerning bacterial compositions, this may be related to the natural properties of Proteobacteria, which are determined by the carbon availability preferences of this phylum [37]. Moreover, Proteobacteria is one of the comparatively young bacterial phyla and has likely evolved in well-developed soils, thus preferring high organic matter contents and high C: N nutrient ratios [38]. It was also proven by a correlation test that the content of soil organic carbon is significantly correlated with the abundance of Proteobacteria (Table S1). Acidobacteria was less abundant when the soil was moister and cooler; thus, the lower soil water content promoted the abundance of Acidobacteria [39] (Figure 3B). In summary, these results suggest that the variation in habitat is accompanied by a change in species abundance, rather than its dominant species at the phylum level on a small regional scale.

Since microorganisms are essential components in providing ecosystem services, microbial community structure can influence a variety of ecosystem processes [6]. Thus, it is of great significance to predict ecological succession by comparing the changes in the soil microbial community in different habitats [40]. In our study, we also noted marked variation in the soil microbial construct among different vegetation types using PCoA and PERMANOVA analysis, which suggested high spatial heterogeneity in soil microbial communities across a small regional scale. Our simultaneous investigation of these microbial groups in replicated, well-separated patches of a variety of ecosystems suggests that different vegetation types have a strong influence on the distributions of the major soil fungi, and relatively little effect on the distributions of the principal bacteria. Similar conclusions were obtained by Jiang et al. [35], who considered that soil fungal communities are more responsive to habitat changes. Similar small-scale heterogeneity and patchy distributions of soil fungal populations have also been observed in other ecosystems [41,42]. There are several possible explanations for this phenomenon. One possibility is that, in addition to shifting the composition and structure of bacterial communities perse [43], the utilization mode of nutrition may cause a difference in community stability. Compared with fungi, which are strongly dependent on the presence of their host, bacteria can metabolize a wider range of compounds, which may explain their relative stability [44]. On the other hand, due to the establishment of a biotrophic relationship between plants and fungi, a previous study proved that fungi form a strong bond with grass, which can make more efficient use of available carbon in plants [45]. We also noticed that the diversity of bacteria and fungi in swamp meadows was lower than that in other grassland types, which may be related to the degree of disturbance of the grassland [46]. Second, the reseeded grassland, steppe meadow, and cultivated grassland were arid grassland types, especially the cultivated grassland used as grazing pasture, in which the original soil was greatly disturbed by human beings. A previous study reported that cultivating grassland caused nutrient and soil outflow from the plant–soil system and a decrease in soil nutrient availability [47]. This may explain why the soil microbial diversity in reseeded grassland, cultivated grassland, and steppe meadow was greater than that in swamp meadow in our study.

Soil pH, soil carbon, and altitude have long been recognized as having a profound influence on the structure and diversity of microbial communities in large-scale areas [48–50]. However, it remains unclear whether the microbial community follows this regular pattern in a small regional area. In our study, RDA showed that the microbial community was

more sensitive to soil properties than to plant characteristics, which is the same as the results of Li et al. [51]. This indicates that the microbial community changes depend on soil rather than plant characteristics. In addition, soil and plant factors have different effects on fungi and bacteria because of the differences in morphological structure and proliferation patterns [52]. In our work, soil organic carbon as an indicator to measure soil nutrients was the most important influencing factor for the microbial community (Figure 9). This result was consistent with the observation that the microbial community was positively correlated with soil SOC [53]. Microbes, especially in the bacterial community, play an important role in nutrient cycling in soils, such as degrading organic matter and organic C from litter mass, which in turn regulates C cycling in soils [54]. Studies have shown that microbial groups are affected by changes in different carbon substrates [55,56], which links microbial functional classification to soil organic matter (SOM) properties. Thus, with the change in soil carbon content in different habitats, the microbial community also changes. In addition, we also noticed that $NO_3^--N$ was the most important factor affecting the bacterial community, which may be related to the function of bacteria. Proteobacteria affect the dominant nitrogen-fixing groups and structures of microorganisms by participating in the soil nitrogen cycle as dominant nitrogen-fixing groups [57]. This result was also confirmed by RDA (Figure 8A), where $NO_3^--N$ was related to the bacterial community structure of the swamp meadow.

For the fungal community, the fungal microbial community composition was strongly related to the plant characteristics (Figure 9B). This was in agreement with the study of [58], which found that the vegetation composition represented a more stable summary of multiple drivers over time. Thus, under different grassland types, it is reasonable that the differences in fungal community composition may also be driven by plant biomass. Furthermore, soil fungi can form symbiotic relationships with plant roots and promote plant growth because fungi have a stronger ability to degrade complex compounds than bacteria [59]. On the other hand, the carbon source of soil fungal growth mainly comes from vegetation litter. Therefore, different plant types will have different effects on the soil microbial structure.

## 5. Conclusions

The soil microbial community can reflect the difference in the plant–soil systems among different grassland types. However, the observed differences were different from the large-scale spatial patterns. From the analysis of the patterns of microbial diversity at small spatial scales, our results indicated that different grassland types have the same dominant species of bacteria and fungi at small spatial scales in the Qilian Mountains, but the relative abundances changed significantly. However, there were significant differences in the construct of the microbial community. The analysis of environmental factors shows that the explanation degree of soil characteristics for the microbial differences was higher than that for the plant characteristics. Among them, organic carbon is a common factor that affects bacterial and fungal communities. Our work highlighted the different patterns of microbial communities at small spatial scales and shed further light on microbial community differences and their main driving factors.

**Supplementary Materials:** The following supporting information can be downloaded at: https://www.mdpi.com/article/10.3390/su14137910/s1, Figure S1: The cladogram shows significant differences between bacterial (A) and fungal (B) enrichment groups. Taxa with significant differences in abundance between different vegetation types are represented by colored dots, and cladogram circles represent phylogenetic taxa from phylum to genus; species with no significant difference are uniformly colored yellow. Only the LDA > 3 for bacteria and >2 for fungi were shown; Table S1: Results of ANOSIM tests comparing pair-wise bacterial and fungal community similarities derived in the matrix (Bray–Curtis distance) for each vegetation type. The ADONIS test examined the significance of separation among different grassland types; Table S2: Correlation relationships between bacterial and fungal phylum with environmental factors.

**Author Contributions:** Investigation, writing—review and editing, W.Z.; data curation and investigation, Y.Y.; data curation and investigation, J.L.; formal analysis and investigation, Y.D. and S.S.; conceptualization, methodology, funding acquisition, data curation, formal analysis, investigation, writing—original draft, writing—review and editing, S.L. All authors have read and agreed to the published version of the manuscript.

**Funding:** This research is funded by the Qinghai Provincial Science and Technology Major Project (2019-SF-A3-1).

**Institutional Review Board Statement:** Not applicable.

**Informed Consent Statement:** Not applicable.

**Data Availability Statement:** Not applicable.

**Acknowledgments:** This work was supported by the Qinghai Provincial Science and Technology Major Project (2019-SF-A3-1). We thank Jingjing Liu, Yiling Dong, and Shifeng Su for their assistance in the field sampling and lab measurement.

**Conflicts of Interest:** The authors declare no conflict of interest.

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
