# Peer review of "Soil Microbial Community Varied with Vegetation Types on a Small Regional Scale of the Qilian Mountains"

_sustainability, doi:10.3390/su14137910_

Round 1
Reviewer 1 Report
Dear authors, overall I think that this paper is interesting but I suggest you some revisions:
1) please rephrase the abstract, it's not clear, especially at lines 17-19.
2) I think that the keywords are not significative
3) I think that the Introduction it's not clear, especially at lines 51-59.
Put a space at line 76 before the citation and I suggest to put the citation [20] immediatly after Curd et al. at line 86
4) I think that the resolution of Figure 1 at page 3 is too poor
5) I don't find the paragraph 2.2 clear
6) I suggest to put "Wang at al." before the citation [27] at line 155
7) In the results, can you explain which are the samples? What ST, RG, SW and CG are standing for?
8) I think that the reference (Fig. 3C) at line 231 it's wrong
I find this first paragraph of the results confused
I suggest to remove the sentence "More information is shown in Fig. 4A, B, C" at line 246 with references at line 245 like this: "[...] Sobs (Fig. 4A), Shannon (Fig. 4B), or Chao1 (FIg. 4C)
9) please check the axes titles in Figure 5 (pag 9)
10) I suggest to the authors to reconsider the Discussion section. You have included so much information and data in the text and seems like you loose the focus of the paper in the overall: I understand what you want to present but, in my opinion, it's very confused
Author Response
1.1 General comments:
1) please rephrase the abstract, it's not clear, especially at lines 17-19.
Response: Thank you for this kind reminder! We had revised the abstract in the manuscript, especially in lines 17-19. Show as follow ”Patterns in soil microbial community distribution can provide important insights into ecological processes, but the patterns of different grassland types remain unclear on small regional scales. Therefore, we characterized and compared the soil microbial communities (n = 4) underlying the four vegetation types in a national natural reserve (reseeding grassland, swamp meadow, steppe meadow, and cultivated grassland) using High-throughput sequencing of the 16S rRNA and ITS. Meanwhile, the plant community and soil physicochemical characteristics were also determined. The results showed that bacterial and fungal communities in all vegetation types have the same dominant species, but the relative abundance differed substantially, which may cause significantly spatial heterogeneities. Specifically, bacteria showed higher variability in different vegetation types than fungi, and the shift in both the bacterial and fungal communities was related more to the soil than to the plant characteristics. Soil organic carbon affected the widest portion of the microbial community, nitrate-nitrogen was the main factor affecting bacteria, and vegetation aboveground biomass was the second-factor affecting fungi. Collectively, these results demonstrate the value of considering multiple small regional spatial scales when studying the relationship between the soil microbial community and environmental characteristics. Our study may have important implications for grassland management following natural disturbances or human alterations.”
2) I think that the keywords are not significative
Response: We emphasize the changes in microbial communities in the small regional area, and we think that the keywords strongly express the important research contents of this article. Therefore, we didn't revise the keywords.
3)I think that the Introduction it's not clear, especially at lines 51-59.
Response: We summarized the distribution pattern and driving mechanism of microorganisms in the large area or even the global area and then put forward the purpose of this study, that is, the study of different vegetation types in the small area of Qinghai-Tibet Plateau needs to be improved. Therefore, we have revised the introduction according to the experts' opinions.
4) I think that the resolution of Figure 1 at page 3 is too poor
Response: We have changed the photo in Figure 1, and the resolution has been improved.
5) I don't find the paragraph 2.2 clear
Response: We have revised the content of paragraph 2.2.Show as follow“We surveyed the vegetation community at the site, and located four similar, well-separated (500–1000 m apart) patches of each vegetation type (August 2019). This study design with replicate plots (rather than replicate plots within apatch) avoids pseudoreplication and makes our results appropriate to landscape-level extrapolation. In each patch, species diversity, abundance, total coverage, and height were investigated in four 50 cm×50 cm quadrats. All aboveground plant parts were collected in each quadrant as the aboveground biomass, which was determined by oven-drying plant samples at 65 °C for 48 h to a constant weight in the laboratory. At the same time, Root samples from a 0–15 cm depth were collected separately using a root drill with a diameter of 7 cm, after which four root drill samples from each plot were mixed to give one composite root sample [27].
Soil samples were collected from quadrats: eight random soil samples were collected from the 0 to 15 cm soil layer in each replicate plot using a soil-drilling sampler (7 cm inner diameter) and combined into one replicate sample; in total, 4 soil samples were obtained in each vegetation type. All soil samples were passed through a 2 mm sieve to remove other materials. Soil samples were immediately sent back to the laboratory in a cooler. Samples were divided into three parts to determine the soil chemical and physical properties and the soil microorganisms. One part of the soil sample was naturally dried in the shade and then sieved through 1 mm mesh for soil chemical analysis, while another part was stored at -80 °C for high-throughput gene detection [28]. A third portion was used in water-stable aggregate analysis.”
6) I suggest to put "Wang at al." before the citation [27] at line 155
Response: We have revised the introduction of the whole article according to the expert's opinion.
7) In the results, can you explain which are the samples? What ST, RG, SW and CG are standing for?
Response: In this study, we discussed the microbial community in different vegetation types. Therefore, four types of vegetation types are samples. In our study, the swamp meadow and steppe meadow are natural grasslands with the dominant species Kobresia humilis and Elymus nutans, respectively. The cultivated grassland was established on the extremely severely degraded alpine meadow for 4 years, mainly with Poa pratensis cv. Qinghai. The reseeding grassland was dominated by E. nutans, which was reseeded on the moderately degraded alpine meadows for 2 years by the reseeding sowing combined operation.
8) I think that the reference (Fig. 3C) at line 231 it's wrong
I find this first paragraph of the results confused
I suggest removing the sentence "More information is shown in Fig. 4A, B, C" at line 246 with references on line 245 like this: "[...] Sobs (Fig. 4A), Shannon (Fig. 4B), or Chao1 (FIg. 4C)
Response: In this study, the first paragraph of the results is the sequencing of soil bacteria and fungi, and we also compared the OTU and phylum levels obtained by sequencing. In addition, according to the expert's opinion, the sentence has been rewritten as follows Sobs (Fig. 4A), Shannon (Fig. 4B), or Chao1 (FIg. 4C).
9) I suggest to the authors to reconsider the Discussion section. You have included so much information and data in the text and seems like you lose the focus of the paper the overall: I understand what you want to present but, in my opinion, it's very confused
Response: We apologize for this misunderstanding! After further consideration, we decide to conduct our discussion in three-part. First, we discussed the microbial composition and draw a conclusion. Secondly, the microbial construct was main discussed. Finally, the relationship was talked over between microbial community and environmental factors.

Reviewer 2 Report
The manuscript "Soil microbial community was related more to soil than plant characteristics on a small regional scale of Qilian Mountain" by Zhao et al. aims to understand the difference in the soil microbial community of alpine grasslands on small regional scales.
As part of their hypothesis, the authors claim to "predicted driving factors from plant and soil characteristics to soil bacterial and fungal communities". However, the information on this topic is reduced in the introduction. Only pH, "strong plant community," altitudinal gradient (the last two not related to this study's aims), and drought index are mentioned.
The authors did a lot of statistical analyses to analyze the differences between the four types of vegetation they studied; however, the presentation is blundering, and the discussion is not very clear.
The studied grasslands were very recently established (2 and 4 years ago, for the reseeding and the cultivated, respectively). However, the authors did not discuss the effect of the previous "moderately degraded" and "extremely severely degraded" alpine meadow effect over the results. There is no discussion of the very small fraction of the variance explained by the multiple soil, plant, and environmental factors analyzed.
The whole discussion section needs to be re-written to highlight the study's novelty.
Some information is missing in the M&M section, while some are repeated from the first in the results.
Specific comments:
L54-55 – This affirmation is not supported by the cited article (please check the cited article, in which the pH had a positive influence indeed over the bacterial community in meadow steppe with grazing)
L74-76 – Why is it relevant in this study the altitudinal gradient influence over the microbial community?
L91-92 Incomplete idea
L128- What do the authors mean by "plant investigation". Please search for a more appropriate subtitle.
L138 – Please describe the diversity index applied.
L143 – Soil sieve was before lab transportation?
L146 – Did the authors mean "air-dried."
L174 - Please describe more details regarding the alpha and beta diversity index.
L174 - The results of the gene abundance were rarified?
L176 – RDA must be described in the Statistical Analyses section
L200 – Please define VPA before first use. Also, apply the abbreviations assigned to the vegetation types (L211, L212).
L205-207 – Please avoid repeat methods.
L214 – OTUs
L219-221 – The authors indicate that their analysis was at an OUT level. Why then is this phrase mention genus and specie level?
L233-234 – This must be specified in the M&M section instead of results.
L243 – Sobs determination was not defined nor mentioned in the M&M section
L266 – The name function is envfit and is usually presented without capital letters and in italics.
L264-266 – The affirmation does not match with the data in Table 1. Please double-check. Be consisting of the determinations are named in full or abbreviated.
L281 – Mantel test? (L303) There is no mention in the M&M section (please indicate the R package used for each analysis used)
L283-284, L307-309 ¬- Please avoid repetition with methods.
L284 – Why, surprisingly?
Figures – please define all the terms used in each figure.
Author Response
Response to Reviewer 2Comments
The manuscript "Soil microbial community was related more to soil than plant characteristics on a small regional scale of Qilian Mountain" by Zhao et al. aims to understand the difference in the soil microbial community of alpine grasslands on small regional scales.
As part of their hypothesis, the authors claim to "predicted driving factors from plant and soil characteristics to soil bacterial and fungal communities". However, the information on this topic is reduced in the introduction. Only pH, "strong plant community," altitudinal gradient (the last two not related to this study's aims), and drought index are mentioned.
The authors did a lot of statistical analyses to analyze the differences between the four types of vegetation they studied; however, the presentation is blundering, and the discussion is not very clear.
The studied grasslands were very recently established (2 and 4 years ago, for the reseeding and the cultivated, respectively). However, the authors did not discuss the effect of the previous "moderately degraded" and "extremely severely degraded" alpine meadow effect over the results. There is no discussion of the very small fraction of the variance explained by the multiple soil, plant, and environmental factors analyzed.
The whole discussion section needs to be re-written to highlight the study's novelty.
Some information is missing in the M&M section, while some are repeated from the first in the results.
Response: We do appreciate all your encouragement and insightful comments below! The whole manuscript has been carefully revised accordingly. Please see our detailed responses.
1)L54-55 – This affirmation is not supported by the cited article (please check the cited article, in which the pH had a positive influence indeed over the bacterial community in meadow steppe with grazing)
2)L74-76 – Why is it relevant in this study the altitudinal gradient influence over the microbial community?
Response: This is great advice, but the aim of citing pH and the altitudinal gradient was to talk about the difference between the research on large and small areas. We have to consider the size of the research area when discussing the microbial community.
3) L128- What do the authors mean by "plant investigation". Please search for a more appropriate subtitle.
Response: We apologize for this misunderstanding! The means of “plant investigation” was investigating the height, cover, and density of all plant species, as well as harvesting the aerial part of the plants. We have made changes to our manuscript.
4) L143 – Soil sieve was before lab transportation?
Response: Yes, we sieve the soil before lab transportation. The soil collected at the field site contains impurities such as stones and plant roots, which may affect the later tests in the lab. From collection to sieving to packaging, we take sterile gloves all the way.
5) L146 – Did the authors mean "air-dried."
Response: According to the requirements of soil chemical properties determination, we had dried the soil used for determination indoors in the dark to ensure that the chemical properties can be truly determined.
6) L174 - The results of the gene abundance were rarified?
Response: The gene abundance in our manuscript is relative, and the whole analysis was carried out by relative abundance.
7)L174 - Please describe more details regarding the alpha and beta diversity index.
Response: We had put details regarding the alpha and beta diversity index in the manuscript, shown as follows” Alpha-diversity metrics (include Chao1 richness estimator, Shannon diversity index were estimated for each sample using the diversity plugin in QIIME2 software.”
8)L176 – RDA must be described in the Statistical Analyses section
Response: We had put the description of RDA in the Statistical Analyses section. show as follow “Redundancy analysis was used to elucidate the relationships between the community and environmental properties using the R vegan package, and the relations were examined using the Monte Carlo permutation”.
9) L200 – Please define VPA before first use. Also, apply the abbreviations assigned to the vegetation types (L211, L212).
Response: Thank you for this suggestion! We have modified it to VPA (Variance Partitioning Analysis) when we first use it. In addition, we also add abbreviations assigned for the vegetation types in each figure and table.
10) L214 – OTUs
Response: OTUs: the final OTU quantity.
11) L219-221 – The authors indicate that their analysis was at an OUT level. Why then is this phrase mention genus and specie level?
Response: In our study, our microbial community difference and diversity were analyzed based on the OTU level, while other analyses are based on the phylum level.
12) L243 – Sobs determination was not defined nor mentioned in the M&M section
Response: Thank you for this suggestion! We had put the determination of Sobs in Methods and materials as follows “Sobs indicates the number of OTU actually measured”.
13) L266 – The name function is envfit and is usually presented without capital letters and in italics.
Response: Thank you for this suggestion! We have revised it according to the expert's opinion.
14) L264-266 – The affirmation does not match with the data in Table 1. Please double-check. Be consisting of the determinations are named in full or abbreviated.
Response: We apologize for this misunderstanding! After further checking, we found that TN was ignored in the manuscript. We had added the description in the manuscript.
15) L281 – Mantel test? (L303) There is no mention in the M&M section (please indicate the R package used for each analysis used)
Response: Thank you for this suggestion! We had put the determination of the Mantel test as follows “We also calculated the Bray-Curtis distance matrix between samples based on the species abundance and the environmental factor, and use mantel test to analyze the correlation between the two distance matrices, using the R vegan package to analyze.”
15)L284 – Why, surprisingly?
Response: Most studies showed that the influence of vegetation community on the microbial community is greater than soil characteristics, our study shows the opposite trend, which surprises us.
16) Figures – please define all the terms used in each figure.
Response: We had defined all the terms used in each figure.

Reviewer 3 Report
Overall, this is an interesting study conducted by Zhou et al., exploring how soil microbial community was related to soil than plant characteristics on a small regional scale of Qilian Mountain. This is a well-written paper and I suggest minor revisions before this manuscript is acceptable for publication. I have clearly indicated all my opinions and suggestions below.
Minor Comments:
Line 101-102: Hypothesis is not clear. Please indicate the basis for your hypothesis.
Line 103-104: Please re-write this sentence.
Line 208: What are ZOTUs? Clustering methods? Why did you use them over OTUs?
Line 223: The figure legend for Fig 2 is confusing and not labelled properly. Please correct.
Line 231: Where is Fig 3C?
Author Response
Response to Reviewer 3Comments
Comments and Suggestions for Authors
Overall, this is an interesting study conducted by Zhou et al., exploring how soil microbial community was related to soil than plant characteristics on a small regional scale of Qilian Mountain. This is a well-written paper and I suggest minor revisions before this manuscript is acceptable for publication. I have clearly indicated all my opinions and suggestions below.
1) Line 101-102: Hypothesis is not clear. Please indicate the basis for your hypothesis.
2) Line 103-104: Please re-write this sentence.
Response: Thank you for this suggestion! We have modified to“We hypothesized that the soil community characterization may vary with vegetation types on a small regional scale. Moreover, determine the key factors that affect the changes in microbial community structure”.
3) Line 208: What are ZOTUs? Clustering methods? Why did you use them over OTUs?
Response: We apologize for this misunderstanding! We want express the final OTU quantity, we have modified it to OTUs. And corresponding OTUs were clustered using the UPARSE pipeline at 97% sequence similarity per sample.
4) Line 223: The figure legend for Fig 2 is confusing and not labelled properly. Please correct.
Response: Figure2 (A), (B) Circos diagram was used to dynamically show the abundance composition of species among each grassland type at phylum classification levels in soil bacterial and fungal communities. (C), (D) The ANOVA of the relative abundance of each dominant species. CG, SW, ST, and RG indicate cultivated grassland, swamp meadow, steppe meadow, and re-seeding grassland, respectively. The figure need not legend.
4) Line 231: Where is Fig 3C?
Response: We apologize for this misunderstanding! That is a wrong writing.

Reviewer 4 Report
Quite interesting research. All the graphs are well-presented. If possible, a graphical abstract may be added for a quick reader.
Author Response
Response to Reviewer 4Comments
Comments and Suggestions for Authors
Quite interesting research. All the graphs are well-presented. If possible, a graphical abstract may be added for a quick reader.
Response: we had added a graphical abstract

Round 2
Reviewer 1 Report
The authors fully responded to the requestes.
Thanks!
Author Response
Responses to comments from reviewers
(Manuscript ID sustainability-1703053)
Dear Prof. Chen,
On behalf of my co-authors, we thank you very much for giving us an opportunity to revise our manuscript, we appreciate editor and reviewers very much for their positive and constructive comments and suggestions on our manuscript entitled “Soil microbial community varied with vegetation types on a small regional scale of Qilian Mountain”. (Manuscript ID sustainability-1703053).
We have studied the reviewer’s comments carefully and have made revisions marked by “Track Changes” function. We have tried our best to revise our manuscript according to the comments. Attached please find the revised version, which we would like to submit for your kind consideration.
We would like to express our great appreciation to you and reviewers for comments on our paper. Looking forward to hearing from you.
Again, thank you!
Wen Zhao
List of Responses
Thank you for your letter and for the reviewers’ comments concerning our manuscript entitled “Soil microbial community varied with vegetation types on a small regional scale of Qilian Mountain”. Those comments are all valuable and very helpful for revising and improving our paper, as well as the important guiding significance to our research. We have studied comments carefully and have made correction which we hope meet with approval. Revised portions are marked in red on the paper. The main corrections in the paper and the responses to the reviewer’s comments are as flowing:
Response to Reviewer 1Comments
Comments and Suggestions for Authors: The authors made an effort to improve the manuscript. However, there are still minor details in order to be published.
1) The presentation of the results is still average. Almost all figures need to be reviewed to avoid overlap between the axes' titles from top to bottom plots. PLEASE do not present P= 0.0000 that does not exist. P<0.001 is appropriate.
Response: Thank you for this kind reminder! We reviewed our figures to avoid overlap between the axes' titles from top to bottom plots. On the other hand, we P value had reviewed in our manuscript.
2) Table 1 and Fig. 6. The presented r, is R os R2 or adjusted R? Please revise.
Response: r value of environmental factors' influence degree (correlation degree with species) analyzed by Envfit test in table 1. The r in the figure.6 represents the correlation between plant and microbial communities.
3) Check Fig. 9B. There is a letter at the bottom of the figure.
Response: Thank you for this kind reminder! We had modified Figure 9B in our manuscript.
4) Please use subscripts and superscripts in figures and especially in the text where corresponds (e.g. NO3, NH4, R2, etc).
Response: We carefully checked and reviewed the subscripts and superscripts in our manuscript.
5) Supplementary material needs to be reviewed as well. Many terms are not described in this material (AGB, BGB, etc).
Response: Supplementary material had been reviewed, we also add description of terms, especially AGB, BGB.

Reviewer 2 Report
The authors made an effort to improve the manuscript. However, there are still minor details in order to be published.
The presentation of the results is still average. Almost all figures need to be reviewed to avoid overlap between the axes' titles from top to bottom plots. PLEASE do not present P= 0.0000 that does not exist. P<0.001 is appropriate.
Table 1 and Fig. 6. The presented r, is R os R2 or adjusted R? Please revise.
Check Fig. 9B. There is a letter at the bottom of the figure.
Please use subscripts and superscripts in figures and especially in the text where corresponds (e.g. NO3, NH4, R2, etc).
Supplementary material needs to be reviewed as well. Many terms are not described in this material (AGB, BGB, etc).
Author Response
Thank you for your letter and for the reviewers’ comments concerning our manuscript entitled “Soil microbial community varied with vegetation types on a small regional scale of Qilian Mountain”. Those comments are all valuable and very helpful for revising and improving our paper, as well as the important guiding significance to our research. We have studied comments carefully and have made correction which we hope meet with approval. Revised portions are marked in red on the paper. The main corrections in the paper and the responses to the reviewer’s comments are as flowing:
Response to Reviewer 1Comments
Comments and Suggestions for Authors: The authors made an effort to improve the manuscript. However, there are still minor details in order to be published.
1) The presentation of the results is still average. Almost all figures need to be reviewed to avoid overlap between the axes' titles from top to bottom plots. PLEASE do not present P= 0.0000 that does not exist. P<0.001 is appropriate.
Response: Thank you for this kind reminder! We reviewed our figures to avoid overlap between the axes' titles from top to bottom plots. On the other hand, we P value had reviewed in our manuscript.
2) Table 1 and Fig. 6. The presented r, is R os R2 or adjusted R? Please revise.
Response: r value of environmental factors' influence degree (correlation degree with species) analyzed by Envfit test in table 1. The r in the figure.6 represents the correlation between plant and microbial communities.
3) Check Fig. 9B. There is a letter at the bottom of the figure.
Response: Thank you for this kind reminder! We had modified Figure 9B in our manuscript.
4) Please use subscripts and superscripts in figures and especially in the text where corresponds (e.g. NO3, NH4, R2, etc).
Response: We carefully checked and reviewed the subscripts and superscripts in our manuscript.
5) Supplementary material needs to be reviewed as well. Many terms are not described in this material (AGB, BGB, etc).
Response: Supplementary material had been reviewed, we also add description of terms, especially AGB, BGB.
